# Sample Complexity of Goal-Conditioned Hierarchical Reinforcement Learning

**Arnaud Robert**
Brain & Behaviour Lab
Dept. of Computing
Imperial College London, UK
a.robert20@imperial.ac.uk

**Ciara Pike-Burke**
Dept. of Mathematics
Imperial College London, UK
c.pike-burke@imperial.ac.uk

**A. Aldo Faisal**
Brain & Behaviour Lab
Depts. of Computing & Bioengineering
Imperial College London, UK
Chair in Digital Health & Data Science
University of Bayreuth, Germany
a.faisal@imperial.ac.uk

## Abstract

Hierarchical Reinforcement Learning (HRL) algorithms can perform planning at multiple levels of abstraction. Empirical results have shown that state or temporal abstractions might significantly improve the sample efficiency of algorithms. Yet, we still do not have a complete understanding of the basis of those efficiency gains, nor any theoretically-grounded design rules. In this paper, we derive a lower bound on the sample complexity for the considered class of goal-conditioned HRL algorithms. The proposed lower bound empowers us to quantify the benefits of hierarchical decomposition and leads to the design of a simple Q-learning-type algorithm that leverages hierarchical decompositions. We empirically validate our theoretical findings by investigating the sample complexity of the proposed hierarchical algorithm on a spectrum of tasks (hierarchical $n$-rooms, Gymnasium's Taxi). The hierarchical $n$-rooms tasks were designed to allow us to dial their complexity over multiple orders of magnitude. Our theory and algorithmic findings provide a step towards answering the foundational question of quantifying the improvement hierarchical decomposition offers over monolithic solutions in reinforcement learning.

## 1   Motivation

Hierarchical Reinforcement Learning (HRL) [27, 8, 9, 4] leverages the hierarchical decomposition of a problem to build algorithms that are more sample efficient. While there is significant empirical evidence that hierarchical implementations can drastically improve the sample efficiency of Reinforcement Learning (RL) algorithms [20, 21, 29, 8], there are also cases where temporal abstraction worsens the empirical sample complexity [16]. Therefore, a natural question to ask is: when does HRL lead to improved sample complexity, and how much of an improvement can it provide?

Theoretical work on sample-complexity bound in Machine Learning has been integral to the development of the field. Moreover, theoretical results (e.g. [7, 18, 3, 15, 26]) often uncover interesting principles useful for improving algorithm design. For example, the Q-learning algorithm analysed in [15] improved our understanding of exploration strategies in model-free RL and the policy gradient

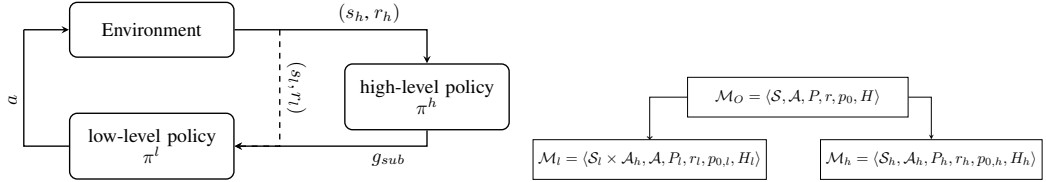

Figure 1: The left block diagram depicts the interactions between the different components of our goal-conditioned hierarchical agent. The diagram suggests that the agent is composed of a low-level policy and a high-level policy that collaborate in order to solve a task. The high-level policy $\pi^h$ observes the pair $(s_h, r_h)$, which denotes the high-level state and reward. It then sends a sub-goal $g_{sub}$ as an input to the low-level policy $\pi^l$. $\pi^l$ observes the pair $(s_l, r_l)$, which encodes the low-level state $s_l$ and the low-level reward $r_l$. To achieve the sub-goal $g_{sub}$, the low-level policy $\pi^l$ interacts with the environment through primitive actions $a$. The right diagram illustrates the decomposition of the original MDP $\mathcal{M}_O$ into the low-level MDP $\mathcal{M}_l$ and the high-level MDP $\mathcal{M}_h$. A detailed description of this decomposition is given in Sec. 2.2

theorem [26] gave birth to a wide range of new RL methods. In contrast, there are few theoretical results in hierarchical RL and many key studies are empirical, e.g. hierarchies of states [8, 10], time [23], or action [28, 22, 2].

To address this gap in the literature, we consider a tabular version of the goal-based approach to HRL [20, 4], and we analyze the induced MDP decomposition to derive a lower bound on the sample complexity of this specific HRL framework. This lower bound allows us to understand when a hierarchical decomposition is beneficial and motivates a new hierarchical Q-learning algorithm that can leverage the hierarchical structure to improve its sample efficiency. In the goal-based HRL framework, a high-level policy and a low-level policy are jointly learned to solve an overarching goal. In such a goal-hierarchical RL system, the high-level policy chooses a sub-goal for the low-level policy, which in turn executes primitive actions to solve the sub-goal (Fig. 1, left diagram). This natural way to break down tasks is universal (i.e., it can be applied to a wide range of tasks) and it induces a decomposition of the original MDP into two sub-MDPs (detailed in Sec. 2.2).

This paper improves our understanding of HRL through the following contributions:

- We provide a lower bound on the sample complexity associated with the hierarchical decomposition (see Sec. 3). This lower bound allows practitioners to quantify the efficiency gain they might obtain from decomposing their task.

- We propose a simple, yet novel, Q-learning-type algorithm for goal-hierarchical RL, inspired by the type of decomposition considered (see Sec. 4).

- We empirically validate the theoretical findings using a synthetic task with hierarchical properties that can be scaled in complexity (see Sec. 5). This evidence confirms that the derived bound is able to successfully identify instances where a hierarchical decomposition could be beneficial (see Sec. 5).

## 2 Background

We consider a system where an agent needs to make a sequence of decisions in an uncertain environment to maximise the sum of cumulated rewards. Such problems are modelled as Markov Decision Processes (MDPs) and can be solved by RL algorithms [25]. When a task is too complex, the number of interactions required to learn a near-optimal policy becomes prohibitive. The task complexity typically depends on the difficulty of temporal credit assignment (which is directly related to the episode length) and the size of the state and action spaces [19]. To address this complexity, HRL leverages temporal abstractions [27] and state abstractions [8] to improve sample efficiency when learning an optimal policy. There exists a wide range of HRL frameworks; see [14] for a survey. In this paper, we focus on the goal-conditioned HRL framework [20, 4]. Of the other HRL frameworks, only the options framework [27] and the resulting semi-Markov Decision Process [12, 30, 5, 11] benefit from some theoretical understanding. However, in practice, the goal-conditioned hierarchical

framework presented in Fig. 1 is often preferred. Unlike the options framework, the goal-conditioned HRL framework requires no prior knowledge about the task [14], and its ability to generalize over the goal space when function approximation is used leads to significant performance gains in benchmark tasks [29, 20, 13]. Existing theoretical work [30, 12, 11] on the options framework does not consider the case where all the hierarchy levels are jointly learned. The regret analysis proposed by [30] focuses on the benefit of leveraging repeating sub-structures in hierarchical MDPs. Other regret analyses [12, 11] highlight the efficiency gain of learning with temporally extended actions (such as options). However, they always assume the set of options is known, and the intra-option policies are not learned. Options are composed of an intra-option policy, which governs the agent's behaviour while the corresponding option is executed, making intra-option policies very similar to the low-level policy we consider. However, the goal-conditioned HRL setting considered in this article quantifies the benefits of state abstraction, action abstraction, and time abstraction while jointly learning all levels of the hierarchy (low-level and high-level policy) through interaction with the environment. A detailed description of the option framework and its connection to HRL is available in Appendix B.

For the remainder of this section, we define episodic finite-horizon MDPs and the hierarchical decomposition we consider.

## 2.1 Episodic Finite-Horizon Markov Decision Process

An episodic finite-horizon MDP is defined by the following tuple: $\mathcal{M} = \langle \mathcal{S}, \mathcal{A}, r, P, p_0, H \rangle$. Where $\mathcal{S}$ is a finite state space of size $|\mathcal{S}|$ and $\mathcal{A}$ is a finite action space of size $|\mathcal{A}|$. The goal of the task is encoded in a terminal state $g \in \mathcal{S}$. We assume the reward function $r(s, g) \in [-a, b]$ (for $a, b \geq 0$) is known $\forall s \in \mathcal{S}, g \in \mathcal{S}$, the reward function penalises each step with a negative reward of at most $-a$ and reward the completion of the task with a positive reward of $b$. The initial state distribution $p_0$ is a distribution over states that is used to determine in which state an episode starts. The learner interacts with the MDP in episodes of at most $H$ time steps. The episode's starting state $s_0 \sim p_0$ is drawn from the initial state distribution. In each time step $t = 0, \ldots, H - 1$, the learner observes a state $s_t$ and chooses an action $a_t$. Given a state action pair $(s_t, a_t)$ the next state $s_{t+1} \sim P(\cdot | s_t, a_t)$ is drawn from the transition kernel. Eventually, the episode ends because the agent reaches the terminal state or has interacted with the environment for H time-steps.

The agent's objective is to select actions that maximize the expected return throughout an episode. We typically assume actions are chosen according to a policy, $a_t \sim \pi(s_t)$, where $\pi$ is a function that maps each state and time step pair to a distribution over actions $\pi : \mathcal{S} \times [H - 1] \to \Delta_{\mathcal{A}}$, and $\Delta_{\mathcal{A}}$ is the set of all probability distributions over $\mathcal{A}$ and $[H]$ is the set of natural numbers up to $H$. The agent aims to select a policy $\pi$ to maximize the sum of expected rewards, $\mathbb{E}[\sum_{t=1}^{H} r_t | a_t \sim \pi(s_t)]$, where the expectation is over the initial state distribution, the policy and the stochastic transitions. Note that it is usually the case for finite-horizon MDPs that the policy also depends on the current time step. However, to simplify notation, we do not make this relation explicit.

For a given policy $\pi$, we define the value function, $V_\tau^\pi(s)$, and the Q-function, $Q_\tau^\pi(s, a)$, at time step $\tau \in [H - 1]$ as follows:

$$V_\tau^\pi(s) = \mathbb{E}\left[ \sum_{t=\tau}^{H-1} r_t | s_\tau = s, a_{\tau:H-1} \sim \pi \right], \quad Q_\tau^\pi(s, a) = \mathbb{E}\left[ \sum_{t=\tau}^{H-1} r_t | s_\tau = s, a_\tau = a, a_{\tau+1:H-1} \sim \pi \right]$$

where $s \in \mathcal{S}$ denotes the state, $a \in \mathcal{A}$ is the action and the notation $a_{\tau:H-1} \sim \pi$ is used to specify that actions between time step $\tau$ and time step $H - 1$ were selected using $\pi$. The optimal policy $\pi^*$ is the policy with the highest value function for every time step and every state, $V_\tau^{\pi^*}(s) = V_\tau^*(s) = \max_\pi V_\tau^\pi(s) \forall \tau \in [H - 1], \forall s \in \mathcal{S}$. There is always a deterministic Markov policy that maximizes the total expected reward in a finite-horizon MDP [24].

In this article, we assess the quality of a policy by its expected value at the beginning of an episode. To lighten the notation, we define $V^\pi = \mathbb{E}_{s_0 \sim p_0}[V_0^\pi(s)]$ to be the expected value from the beginning of an episode where the expectation is taken over initial states.

## 2.2 Episodic Finite-Horizon Hierarchical MDP

For a given episodic finite-horizon MDP $\mathcal{M}_o$, we assume it can be hierarchically decomposed into a pair of MDPs $(\mathcal{M}_l, \mathcal{M}_h)$ as illustrated on right diagram of Fig. 1. To avoid ambiguity, we use

the following notation: the subscript $o$ denotes the original MDP, while subscripts $l$ and $h$ denote low-level and high-level MDPs, respectively.

The low-level and high-level MDPs consist of the following tuples $\mathcal{M}_l = \langle \mathcal{S}_l \times \mathcal{A}_h, \mathcal{A}, r_l, P_l, p_{0,l}, H_l \rangle$ and $\mathcal{M}_h = \langle \mathcal{S}_h, \mathcal{A}_h, r_h, P_h, p_{0,h}, H_h \rangle$, respectively. To be a valid hierarchical decomposition, we require that these MDPs satisfy the following set of conditions:

**Action space:** The low-level action space consists of the set of primitive actions that the agent can use to interact with the environment. It is equivalent to the original MDP action space $\mathcal{A}$. The high-level action space $\mathcal{A}_h$ is the set of the sub-goals the high-level agent can instruct to the low-level agent. We assume that the set of sub-goals encoded in $\mathcal{A}_h$ is sufficient to solve the task for any state. Note that the set of available actions $\mathcal{A}_h$ depends on the current high-level state $s_h$. To simplify our notation, we do not make this relationship explicit.

**State spaces:** The low-level state $s_l$ and the high-level state $s_h$ contain all necessary information to reconstruct the corresponding state, $s$, in the original MDP. States $s \in \mathcal{S} \subset \mathbb{R}^d$ are usually described as multi-dimensional vectors, where each dimension encodes a specific characteristic. For example, a state description can be factored in a tuple $(s_l, s_h) \in \mathcal{S}_l \times \mathcal{S}_h$ with a part of the state description that belongs to the low-level MDP and another part to the high-level MDP. Hence, in this work, we consider that any state $s \in \mathcal{S}_o$ can be represented by a tuple $(s_l, s_h) \in \mathcal{S}_l \times \mathcal{S}_h$. Additionally, since the low-level policy is goal-conditioned, its state space also contains the goal description leading to the following state space for the low-level MDP: $\mathcal{S}_l \times \mathcal{A}_h$, a complete low-level state consists of the concatenation of the low-level state description $s_l$ and the sub-goal description $a_h$.

**Initial state distribution:** The high-level initial state distribution $p_{0,h}$ is a restriction of the original state distribution $p_0$ on $\mathcal{S}_h$. The low-level initial state distribution $p_{0,l}(\cdot|s_{h,0})$ is conditioned on the initial high-level state $s_{h,0}$ and spans the low-level space, ensuring that $p_0(s) = p_{0,h}(s_h)p_{0,l}(s_l|s_h)$, where $s_l$ and $s_h$ are the decomposition of $s$.

**Transition functions:** The low-level transition function $P_l$ is the restriction of $P$ on $\mathcal{S}_l \times \mathcal{A}_h$. One challenge in HRL is that the high-level transition function, $P_h$, depends on the low-level policy since the quality of the low-level policy influences the likelihood of reaching a sub-goal state. The high-level transition probability $P_h(s'_h|s_h, a_h, \pi_l)$ is the probability that the agent transitions to $s'_h$ given the current high-level state $s_h$, the sub-goal $a_h$ and low level policy $\pi_l$. Since $P_h$ depends on the low-level policy, it is non-stationary, making the learning task more challenging.

**Reward functions:** Since the terminal states for the original MDP belong to $\mathcal{S}$ and the sub-goals for the low-level MDP lie in $\mathcal{S}_l$ the low-level reward function can be obtained from the original reward function, $r_l(s_l, g_{sub}) = 2r(s, g)$, where $s$ and $g$ are the reconstruction of the low-level state and the sub-goal in the original MDP, using the current high-level state. The high-level reward function is the sum of rewards obtained by the low level during the sub-episode, where the high-level action plays the role of a sub-goal: $r_h(s_h, a_h) = \sum_{t=1}^{H_l} r_l(s_{l,t}, a_h)$.

**Horizons:** The original MDP allows an episode to last at most $H$ steps. Consequently, the horizons of the high-level, $H_h$, and low-level, $H_l$, MDPs must satisfy the following equality $H = H_h H_l$.

Note that we can always find a decomposition that satisfies these assumptions; a naive way to decompose any MDP would be to consider a high-level agent whose only action encodes the end goal of the task and a low-level with complete state information (i.e. it does not use state abstraction). While this decomposition is valid, it is not necessarily beneficial. Here, our goal is to identify when a given decomposition is useful, specifically in terms of improvements in the sample efficiency.

We denote by $\pi_l$ a policy interacting with the low-level MDP $\mathcal{M}_l$, and $\pi_h$ a policy interacting with the high-level MDP $\mathcal{M}_h$. In goal-conditioned HRL, the low-level policy maps a low-level state and sub-goal pair to an action: $\pi_l : \mathcal{S}_l \times \mathcal{A}_h \rightarrow \mathcal{A}_l$ and the high-level policy maps a high-level state to a high-level action: $\pi_h : \mathcal{S}_h \rightarrow \mathcal{A}_h$. Each policy can be evaluated using the corresponding high and low-level value functions $V_l^{\pi_l}$ and $V_h^{\pi_h}$. Similar to the non-hierarchical case, we can define optimal high-level and low-level policies as $\pi_l^* = \text{argmax}_{\pi_l} V_l^{\pi_l}$ for the low-level policy and $\pi_l^* = \text{argmax}_{\pi_h} V_h^{\pi_h}$ for the high-level policy. Moreover, as shown below, every pair of policies $(\pi_l, \pi_h)$ can be combined to produce a policy $\pi$ that interacts with the original MDP $\mathcal{M}_o$.

**Definition 2.1.** A hierarchical policy consists of a pair $(\pi_l, \pi_h)$ that can be mapped to a policy $\pi$ in the original MDP $\mathcal{M}_o$ as follows:

$$\pi(a|s) = \pi(a|s_l, s_h) = \sum_{a_h \in \mathcal{A}_h} \pi_h(a_h|s_h)\pi_l(a|a_h, s_l). \tag{1}$$

The optimal hierarchical policy is obtained when merging $(\pi_l^*, \pi_h^*)$. It is important to note that not all policies $\pi$ in the original MDP have a corresponding decomposition $(\pi_l, \pi_h)$, and in particular, there is no guarantee that the optimal policy in the original MDP can be decomposed.

We aim to understand when a hierarchical decomposition of the MDP allows us to learn a near-optimal policy faster. Therefore, we are interested in evaluating the performance of the combination of $\pi_l$ and $\pi_h$ while they interact with the original MDP $\mathcal{M}_o$. To convey the fact that we are evaluating a hierarchical policy in the original MDP, we use the following notation: given a pair of policies $(\pi_l, \pi_h)$ and their associated policy in the original MDP, $\pi$, the value function of the hierarchical policy is denoted by $V_o^{\pi_l, \pi_h} = \mathbb{E}_{s_0 \sim p_0}[V_{o,0}^{\pi}(s_0)]$, where the subscript $o$ is a reminder that we are evaluating a policy on the original MDP $\mathcal{M}_o$.

When learning in a decomposed MDP, the learner has to learn two policies, the high-level policy, $\pi_h$, and the low-level policy, $\pi_l$. This is done in an episodic setting where an episode unfolds as follows. Firstly, the learner observes the initial state and uses the high-level policy to find the most appropriate sub-goal. For the next $H_l$ time steps, the low-level policy attempts to solve the sub-goal. The low-level agent updates its policy at the end of each low-level step. Once the $H_l$ time steps are over or if the sub-goal has been reached, the high-level agent observes a new high-level state and can finally perform an update to its policy. The high-level agent instructs a new sub-goal if the overall task is not completed. These interactions are repeated until the task is completed or the horizon $H$ is reached. We can now think of HRL as two agents interacting with the environment. Often, each agent will try to find the policy that maximizes their value function, $\max_{\pi_l} V_l^{\pi_l}$ and $\max_{\pi_h} V_h^{\pi_h}$.

### 2.3 Probably-Approximately Correct RL

We aim to find, in as few episodes as possible, a pair of policies $(\pi_l, \pi_h)$ with a near-optimal value. To formalize this, we introduce the Probably-Approximately Correct (PAC) RL notion. We denote by $\Delta_k$ the sub-optimality gap, that is the difference between the optimal (non-hierarchical) policy $\pi^*$ and the current hierarchical policy $(\pi_l^k, \pi_h^k)$: $\Delta_k := V_o^* - V_o^{\pi_l^k, \pi_h^k}$. Note that both policies are evaluated on the original MDP $\mathcal{M}_o$. The PAC guarantee in this paper follows the definition in [6].

**Definition 2.2.** An algorithm satisfies a PAC bound $N$ if, for a given input $\epsilon, \delta > 0$, it satisfies the following condition for any episodic fixed-horizon MDP: with probability at least $1 - \delta$, the algorithm plays policies that are at least $\epsilon$-optimal after at most $N$ episodes. That is, with probability at least $1 - \delta$, $\max\{k \in \mathbb{N} : \Delta_k > \epsilon\} \leq N$, where $N$ is a polynomial that can depend on the properties of the problem instance.

In Section 3, we will bound the sample complexity of HRL algorithms. In this context, the sample complexity refers to the number of episodes, $N$, in the original MDP, during which the algorithm may not follow a policy that is at least $\epsilon$-optimal with probability at least $1 - \delta$.

### 2.4 Running Example

We consider the following companion example. The original MDP describes the task of solving a maze in a grid-world environment. The state consists of a tuple $(R, C)$ that indicates in which room, $R$, and which cell within that room, $C$, the agent is currently in. The reward function incurs a small cost, $-a$, at each time step unless the agent reaches the absorbing goal state. Once the goal state is reached, the agent stops receiving penalties and receives a reward of $0$ for all the remaining time steps. Mathematically, $r(s) = -a\mathbb{1}\{s \neq g\}$ where $g \in \mathcal{S}$ is the goal state, and $\mathbb{1}$ is the indicator function.

We can decompose this MDP as follows. The high-level MDP describes a similar maze, but instead of moving from cell to cell, the agent moves from room to room, so the state is just the current room. The high-level agent aims to find the room sequence that leads to the goal. Hence, at each (high-level) time step, it indicates the most valuable exit the low-level agent should take from the room. As specified in Section 2.2, the high-level reward for a sub-goal is the sum of the rewards accumulated by the low-level agent during that sub-episode. The low-level agent is myopic to other rooms - it only sees the current room and the exit it has to reach, and it receives a penalty of $-2a$ for each action it takes unless it reaches the sub-goal, in which case it does not receive any penalty. Hence, if $g_{sub}$ is the sub-goal, it receives reward $r(s) = -2a\mathbb{1}\{s \neq g_{sub}\}$.

We will return to this example throughout the paper, but it should be noted that the framework we consider is general enough to be applied to a wide range of tasks. One such example is robotics, where

the low-level agent would be tasked with controlling the joints of the robot to produce movements selected by the high-level policy, whose goal is to perform tasks that require a sequence of distinct movements (i.e. navigational tasks, manipulation tasks or a combination of both).

## 3 Lower Bound on the Sample Complexity of HRL

It has been proven in [7] that, for any RL algorithm, the number of sample episodes necessary to obtain an $(\epsilon, \delta)$-accurate policy (in the original MDP) is lower bounded by:

$$\mathbb{E}[N] = \Omega\left(\frac{|\mathcal{S}||\mathcal{A}|H^2}{\epsilon^2}\ln\left(\frac{1}{\delta+c}\right)\right), \tag{2}$$

where $c$ is a positive constant.

We now extend this result to hierarchical MDPs. Before doing so, it is essential to notice that even the best hierarchical policy (as constructed in Eq. (1)) might be sub-optimal. This is a direct consequence of the goal-conditioned architecture. If, while executing a sub-episode, it appears that another sub-goal becomes more valuable, the architecture proposed does not allow interruptions. The agent will first have to complete the current sub-episode before being able to adapt to the new circumstances. Let $V_o^{\pi_l^*, \pi_h^*}$ denote the value of the optimal hierarchical policy value function in the original MDP. Then, the sub-optimality gap is larger than the gap between the current policy pair and the optimal hierarchical policy $\Delta_k = V_o^* - V_o^{\pi_l^k, \pi_h^k} \geq V_o^{\pi_l^*, \pi_h^*} - V_o^{\pi_l^k, \pi_h^k}$. Therefore, if for some $N$, $V_o^{\pi_l^*, \pi_h^*} - V_o^{\pi_l^k, \pi_h^k} \geq \epsilon$ for at least $N$ episodes, it must also be the case that $\Delta_k \geq \epsilon$ for at least $N$ episodes. Hence, $N$ is a lower bound on the number of episodes where the algorithm must follow a sub-optimal policy.

In the following theorem, we lower bound the number of episodes required to learn a pair of policies $(\pi_l, \pi_h)$ which are $\epsilon$-accurate with respect to the optimal hierarchical policy $(\pi_l^*, \pi_h^*)$. By the above argument, this will also be a lower bound on the number of episodes necessary to learn an $\epsilon$-accurate policy with respect to the optimal policy $\pi^*$.

**Theorem 3.1.** *There exist positive constants $c_l$, $c_h$ and $\delta_0$ such that for every $\delta \in (0, \delta_0)$ and for every algorithm $A$ that satisfies a PAC guarantee for $(\epsilon, \delta)$ and outputs a deterministic policy, there is a fixed horizon MDP such that $A$ must interact for*

$$\mathbb{E}[N] = \Omega\left(\max\left(\frac{|\mathcal{S}_l||\mathcal{A}_h||\mathcal{A}|H_l^2}{\epsilon^2}\ln\left(\frac{1}{\delta+c_l}\right), \frac{|\mathcal{S}_h||\mathcal{A}_h|H_h^2}{\epsilon^2}\ln\left(\frac{1}{\delta+c_h}\right)\right)\right) \tag{3}$$

*episodes, in the original MDP, until the policy is $(\epsilon, \delta)$-accurate.*

The complete proof is in Appendix A.1. In the following, we highlight the main steps.

**Sketch of the proof:** An $\epsilon$-accurate pair of policies must satisfy the following inequality, $|V_o^{\pi_l^*, \pi_h^*} - V_o^{\pi_l, \pi_h}| \leq \epsilon$. To find a lower bound on the number of episodes $N$ before we obtain an $\epsilon$-accurate pair of policies $(\pi_l, \pi_h)$ we used the following steps:

(i) We decompose the objective using the triangle inequality, $|V_o^{\pi_l^*, \pi_h^*} - V_o^{\pi_l^*, \pi_h}| + |V_o^{\pi_l^*, \pi_h} - V_o^{\pi_l, \pi_h}| \leq \epsilon$.

(ii) We show that the number of samples required to guarantee $|V_o^{\pi_l^*, \pi_h^*} - V_o^{\pi_l^*, \pi_h}| \leq \epsilon/2$ is bounded by $\Omega\left(\frac{|\mathcal{S}_h||\mathcal{A}_h|H_h^2}{\epsilon^2}\ln\left(\frac{1}{\delta+c_h}\right)\right)$

(iii) We show that the number of samples required to guarantee $|V_o^{\pi_l^*, \pi_h} - V_o^{\pi_l, \pi_h}| \leq \epsilon/2$ is bounded by $\Omega\left(\frac{|\mathcal{S}_l||\mathcal{A}_H||\mathcal{A}|H_l^2}{\epsilon^2}\ln\left(\frac{1}{\delta+c_l}\right)\right)$

Combining these three steps gives us the result in Theorem 3.1; see A.1 for more details.

### 3.1 Interpretation of the Sample Complexity Bound:

By comparing this lower bound[1] to that in the original MDP, we can identify the problem characteristics that might lead to improved sample efficiency. In general, only one of the two MDP characteristics will dominate the overall sample complexity because of the *max* operator in the bound in Eq. 3. To maintain the *max* as small as possible, the complexity should be distributed between the low- and high-level MDP as evenly as possible. We discuss some of these key insights below:

**State abstraction:** Only one of the two-state space cardinalities will dominate the bound in Eq. 3. This suggests that an efficient decomposition tends to separate the original state space as evenly as possible between the two levels of the hierarchy. Another phenomenon at stake is the low-level re-usability. Due to the state abstraction, the low-level agent can re-use its learned policy in different states (i.e. different states $s_1, s_2 \in \mathcal{S}$ whose low-level component $s_l$ are the same). We rewrite the lower bound 3 in terms of the *re-usability index* $\kappa = \frac{|\mathcal{S}|}{|\mathcal{S}_l|}$.

$$\mathbb{E}[N] = \Omega\bigg( \max \Big( \frac{\frac{|\mathcal{S} \times \mathcal{A}_h|}{\kappa}|\mathcal{A}|H_l^2}{\epsilon^2} \ln\Big(\frac{1}{\delta + c_l}\Big), \frac{|\mathcal{S}_h||\mathcal{A}_h|H_h^2}{\epsilon^2} \ln\Big(\frac{1}{\delta + c_h}\Big)\Big)\bigg). \qquad (4)$$

Eq. 4 highlights that a large re-usability index improves the sample efficiency.

**Temporal abstraction:** Similarly, only one of the two-time horizons will dominate the bound, again suggesting a fair repartition of the load. The temporal abstraction (reducing $H$ to $H_h$ and $H_l$) simplifies the credit assignment problem for the high-level and the low-level policies by giving denser feedback. The low-level agent is rewarded for completing sub-tasks that are significantly shorter than the original task, and the high-level trajectory consists of significantly fewer (high-level) steps than a trajectory in the original MDP.

**High-level action space:** This is the only term that appears on both sides of the $\max(\cdot, \cdot)$ in Eq. 3. This highlights that both the high-level and the low-level benefit from a compact sub-goal representation.

It is interesting to note the contrast between the state space decomposition and the design of the high-level action space. To find efficient state decomposition, the amount of information available at each level must be distributed among each level of the hierarchy. In the case of the sub-goal space, it appears that both levels benefit from a compact representation.

The above discussion highlights properties of the hierarchical decomposition that could improve sample complexity. Note, however, that our bound also shows that a hierarchical decomposition does not always improve the sample efficiency. Indeed, there will be some settings where using a "bad" hierarchical decomposition does not improve the sample complexity. Our bound can, therefore, provide a sanity check to determine whether a hierarchical decomposition *could* lead to an improved sample complexity. However, finding an algorithm that achieves this improved sample complexity can still be challenging. Nevertheless, the proposed Q-learning-based hierarchical algorithm empirically demonstrates the potential benefits of leveraging the considered decomposition. In Section 5, we consider several MDP decompositions and empirically validate that when our bound suggests the hierarchical decomposition is beneficial, our algorithm (see Sec. 4) leverages this to achieve lower sample complexity.

## 4 Stationary Hierarchical Q-Learning

Once we know that we are in an MDP where the hierarchical decomposition could lead to improved sample complexity, the next challenge is to design an algorithm to exploit this. This section proposes the *Stationary Hierarchical Q-learning* algorithm (SHQL) for this purpose. One of the most challenging aspects of jointly learning a pair of policies is the non-stationarity of the high-level transition dynamics, $P_h$. It was briefly mentioned (in Sec. 2.2) that the high-level transition function, $P_h$, is non-stationary since it depends on the low-level policy, $\pi_l$ with the next high-level state depending on whether $\pi_l$ managed to reach the sub-goal. To address this issue, we leverage the fact that the algorithm knows what a successful sub-episode is, i.e. it knows if the low-level agent managed to

---

[1]Note that this is a lower bound - we still do not know if there exist algorithms which achieve this lower bound.

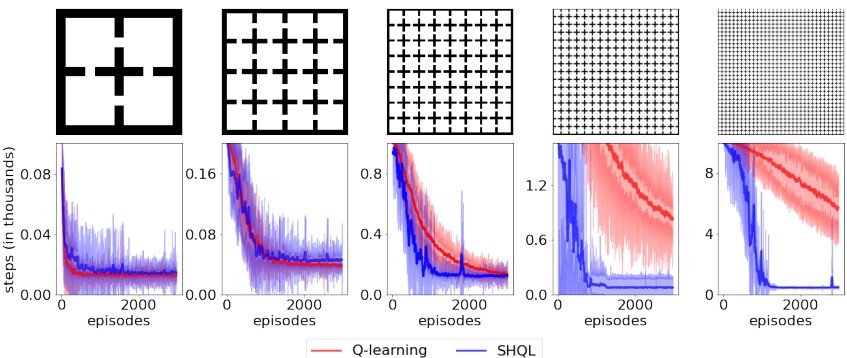

Figure 2: The grid of plots depicts, on the top row, the mazes whose size ranges from 4 rooms to 1024 rooms. The bottom row shows the steps required for SHQL (in blue) and Q-learning (in red) to complete the maze. The standard deviation is obtained by running ten different seeds.

arrive at the desired sub-goal. In the case of off-policy methods, this is typically leveraged with hindsight correction [17]. However, in the considered on-policy setting, we propose that the algorithm only makes an update if the low-level agent behaves reasonably well (i.e. solving the sub-goal). In this way, the algorithm filters all bad examples from the training set, and the behaviour of $P_h$ is more stable. Note, however, that the reward function of the high-level agent remains non-stationary. At first, sub-goals won't be solved optimally, incurring a small reward to the high-level agent. However, the associated reward will increase as the low-level agent learns to solve sub-goals more efficiently. As detailed in the function *LowLevelUpdate* in Algorithm 1, the low-level agent performs Q-learning updates on the observed low-level transitions and rewards. The high-level agent also performs Q-learning updates, but only on successful transitions, as specified at line 10 of Algorithm 1.

**Algorithm 1:** Stationary Hierarchical Q-learning (SHQL)

---

**Input:** $Q^l_{:,:,:} = 0$, $Q^h_{:,:} = 0$, $done_h = False$,
$\quad\quad t = k = 0$

1 **while** *not $done_h$ and $k < K$* **do**
2 $\quad$ Observe $s_h, s_l$
3 $\quad$ $g_{sub} = \pi_h(s_h)$
4 $\quad$ $done_l = False$ /* True if sub-goal solved */
5
6 $\quad$ **while** *not $done_l$ and $t < T$* **do**
7 $\quad\quad$ $a_l = \pi_l(s_l)$
8 $\quad\quad$ Observe $s'_l, r_l, done_l$
9 $\quad\quad$ *LowLevelUpdate*$((s_l, a_l, r_l, s'_l, g_{sub}))$
10 $\quad\quad$ $s_l = s'_l$
11 $\quad\quad$ $t = t + 1$
12 $\quad$ Observe $s'_h, r_h, done_h$
13 $\quad$ **if** *$done_l$* **then**
14 $\quad\quad$ $Q^h_{nxt} = \max\limits_{g} Q^h_{s'_h,g}$
15 $\quad\quad$ $Q^h_{s_h,g_{sub}} = Q^h_{s_h,g_{sub}} + \alpha * (r_h + Q^h_{nxt})$
16 $\quad$ $s_h = s'_h$
17 $\quad$ $k = k + 1$
18 **Function** LowLevelUpdate(*s, a, r, s' $g_{sub}$*):
19 $\quad$ $Q^l_{nxt} = \max\limits_{a'} Q^l_{g_{sub},s',a'}$
20 $\quad$ $Q^l_{g_{sub},s,a} = Q^l_{g_{sub},s,a} + \alpha * (r + Q^l_{nxt})$
21 $\quad$ **return** $Q^l$

## 5   Experiments

We now empirically evaluate[2] the impact of the decomposition on various MDPs to validate the lower bound found in Section 3 and evaluate the performance of our proposed SHQL algorithm. To satisfy the assumption of hierarchical structure, the environments considered are a generalization of the *four-room* problem with an arbitrary number of rooms, called the $n$-rooms problem. The entire maze is built by arranging an arbitrary number of rooms on a grid. The high-level task would involve

---

[2]Experiments were run on a $12^{th}$ Gen Intel Core i7 with 16GB of RAM, to train the agents on the largest maze considered takes $\sim 7$ minutes.

learning the shortest sequence of rooms that lead the agent from the starting position (the top left room) to the goal room (the bottom right room). The low-level task is to learn how to navigate within each room and to reach the instructed hallway. To further modulate the task's difficulty (in addition to the maze size), we vary the room profiles used, as depicted in the rightmost plot of Fig. 4.

The set of MDPs generated by these environments are the following:
**The original MDP:** This is a standard grid-world MDP, where the state space indicates the cell where the agent is located, and the action space allows the agent to move one cell in any cardinal direction (North, South, East, West). To obtain stochastic environments, each action has a success probability of $p_{success} = 4/5$. In case of failure, the action will be chosen at random.
**The high-level MDP:** The high-level state space is restricted to the room where the agent is currently located, and the exact position of the agent within that room is abstracted away. The high-level actions instruct the low-level to reach one of the available hallways. Note that not all rooms have access to the four hallways.
**The low-level MDP:** The low-level agent only observes the agent's current location within a room and the goal instructed by the high-level agent (one of the reachable hallways). It then uses the primitive action space (the four cardinal directions) to reach the desired hallway. All required code to reproduce the experiments is made available online [1].

## 5.1 Identical Rooms

We first introduce the experimental setting in its simplest form. The environments considered in this subsection are mazes built by assembling identical rooms without obstacles (i.e. the top room profile in Fig. 4). Fig. 2 illustrates the empirical performance of our SHQL algorithm against Q-learning in the original MDP. As expected for simple mazes (e.g. with 4 or 16 rooms), the hierarchical decomposition does not provide much improvement. Still, as the problems grow more complex, the empirical evaluation suggests a significant improvement in sample efficiency. This is also confirmed by our bound (yellow curve on the rightmost plot of Fig. 4), which highlights that the efficiency gain of HRL is mostly achievable in complex MDPs (i.e. MDPs with large state and action spaces). It is essential to notice that in this experiment, the low-level decomposition remains constant for a given set of room profiles. This is why the benefit of HRL increases with the number of rooms (i.e. the high-level state space) until a plateau is reached. Once the bound is dominated by the high-level MDP, the unchanging complexity of the low-level MDP causes the ratio between the RL bound (Eq. 2) and the high-level part of the HRL bound (Eq. 3), $\frac{|\mathcal{S}||\mathcal{A}|H}{|\mathcal{S}_h||\mathcal{A}_h|H_h}$, to remain constant (even though number of rooms might still grow).

## 5.2 Different $n$-rooms & Gymansium Taxi Task

To make the task more challenging, we next increase the number of room profiles used to construct the mazes. As depicted in the rightmost plot of Fig. 4 we considered four different room profiles, each one with a different obstacle in the room. The low-level agent must now learn to navigate multiple

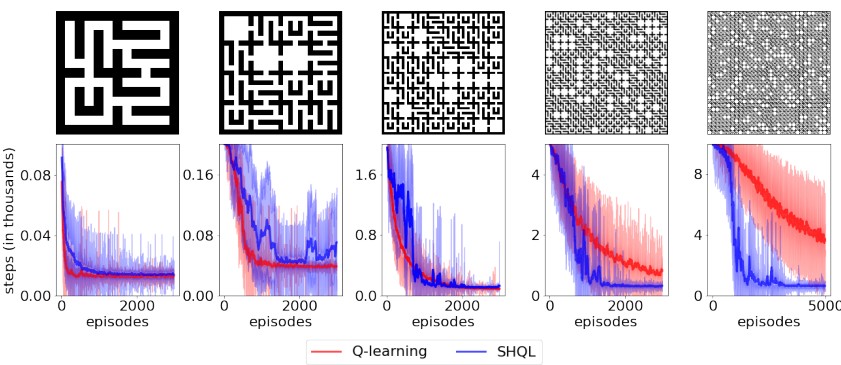

Figure 3: Those plots are similar to the ones shown in Fig. 2, showing the performance obtained on mazes built from four different room layouts.

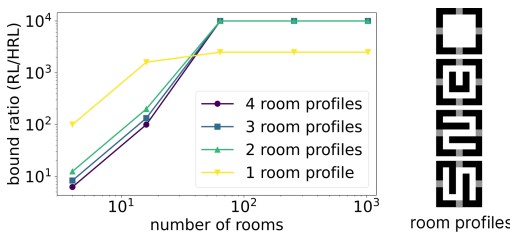

Figure 4: The left-hand plot shows the evolution of the ratio between the RL bound Eq. (2) and the HRL bound Eq. (3) for various mazes and different room profiles. The plateau is obtained when the high-level MDP dominates the bound, leading to the following ratio: $\frac{|\mathcal{S}||\mathcal{A}|H}{|\mathcal{S}_h||\mathcal{A}_h|H_h}$. The curves are colour-coded such that a darker curve indicates more room profiles were considered. The right-hand side of the plot shows the different room profiles available to build the mazes.

types of rooms to reach the sub-goal sent by the high-level agent. The performance of the algorithms with four different rooms is shown in Fig. 3. The introduction of different room profiles allows us to modulate the complexity of the low-level MDP, in contrast to varying the number of rooms, which only affects the complexity of the high-level MDP. This additional complexity results in a larger state space $\mathcal{S}_l$ but may also result in a longer horizon $H_l$ as the optimal trajectory might require more time to navigate around obstacles to reach the desired hallway successfully. It also becomes evident that, as the number of rooms increases, the hierarchy's benefits become more significant. Nevertheless, comparing Fig. 2 and Fig. 3 we can observe that the introduction of various room layouts has little effect on the Q-learning curve (in red). At the same time, it makes the task slightly more challenging for the HSQL learning curve (in blue), especially when the number of rooms is small since it suffers from the increased complexity of the low-level MDP. But, when the number of rooms is sufficiently large for the high-level MDP complexity to dominate the bound, the benefit of hierarchical decomposition becomes evident. The evolution of the bound ratio (HRL/RL) for the various MDPs considered is shown in the leftmost plot of Fig. 4. It shows that the low-level MDP dominates the bound when the maze consists of a small number of rooms. However, the curves clearly indicate that the expected sample efficiency improves as the high-level MDP becomes more complex (i.e., balancing the complexity between the two levels of the hierarchy). This result is also supported by empirical evidence as illustrated in Figs. 2, 3, 5, and 6. The Gymnasium Taxi environment [9] experiments presented in Appendix A.2.2 further validate our approach and conclusion on an entirely different task.

## 6  Conclusion

In this work, we analysed the sample complexity of goal-conditioned HRL. To the best of our knowledge, we provide the first result that analyses the decomposition induced by goal-conditioned HRL. In particular, our lower bound offers a valuable tool for practitioners that could help them decide whether they should consider a hierarchical decomposition for their problem. We also designed a novel algorithm that can leverage the hierarchy to improve its sample efficiency and implemented this on a set of hierarchical tasks. These experimental results further emphasizes the usefulness of the proposed bound since our theoretical findings support empirical efficiency gains.

Although this paper has taken a significant first step in bettering our understanding of the benefits of hierarchical decomposition, there is still scope for further work in this area. Three immediate open questions are: (i) whether our lower bound could be refined by explicitly accounting for the interactions between the low-level and the high-level agent, (ii) is it possible to design an algorithm that can theoretically match the proposed lower bound, (iii) the current results only consider the cardinality of the sub-goal space because we assumed that the sub-goal-space was given and that all sub-goals where solvable. Methods that design efficient sub-goal spaces remain largely unexplored and are a critical aspect of the design of HRL algorithms. Moreover, the insights we proposed are framed in a tabular setting and do not yet extend to a continuous setting where function approximation could be leveraged to allow the low-level agent to generalise over sub-goals and to consider setting beyond the tabular case described in this article. Overcoming those limitations is an interesting direction for future work.

## Acknowledgments

AR was supported by an EPSRC CASE studentship supported by Shell, and AAF was supported by a UKRI Turing AI Fellowship (EP/V025449/1).

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

# A Appendix

## A.1 Proof of Theorem

Theorem 3.1 states that there exist positive constants $c_l$, $c_h$ and $\delta_0$ such that for every $\delta \in (0, \delta_0)$ and for every algorithm $A$ that satisfies a PAC guarantee for $(\epsilon, \delta)$ and outputs a deterministic policy, there is a fixed horizon MDP such that $A$ must collect

$$\mathbb{E}[N_e] = \Omega\left( \max \left( \frac{|\mathcal{S}_l||\mathcal{A}_h||\mathcal{A}|H_l^2}{\epsilon^2} \ln\left(\frac{1}{\delta + c_l}\right), \frac{|\mathcal{S}_h||\mathcal{A}_h|H_h^2}{\epsilon^2} \ln\left(\frac{1}{\delta + c_h}\right) \right) \right) \tag{5}$$

episodes until its policy is $(\epsilon, \delta)$-accurate.

*Proof.* An $\epsilon$-accurate pair of policies $(\pi_l, \pi_h)$ satisfies
$|V_o^{\pi_l^*, \pi_h^*} - V_o^{\pi_l, \pi_h}| \leq \epsilon$. Note that by the triangle inequality, if $|V_o^{\pi_l^*, \pi_h^*} - V_o^{\pi_l^*, \pi_h}| + |V_o^{\pi_l^*, \pi_h} - V_o^{\pi_l, \pi_h}| \leq \epsilon$, then we will have $|V_o^{\pi_l^*, \pi_h^*} - V_o^{\pi_l, \pi_h}| \leq \epsilon$. We therefore focus on showing:

(i) the number of samples required to guarantee $|V_o^{\pi_l^*, \pi_h^*} - V_o^{\pi_l^*, \pi_h}| \leq \epsilon/2$ is bounded by
$\Omega\left( \frac{|\mathcal{S}_h||\mathcal{A}_h|H_h^2}{\epsilon^2} \ln\left(\frac{1}{\delta + c_h}\right) \right)$

(ii) the number of samples required to guarantee $|V_o^{\pi_l^*, \pi_h} - V_o^{\pi_l, \pi_h}| \leq \epsilon/2$ is bounded by
$\Omega\left( \frac{|\mathcal{S}_l||\mathcal{A}_H||\mathcal{A}|H_l^2}{\epsilon^2} \ln\left(\frac{1}{\delta + c_l}\right) \right)$

Then, once we have both (i) and (ii), we know that after

$$\Omega\left( \max \left( \frac{|\mathcal{S}_l||\mathcal{A}_h||\mathcal{A}|H_l^2}{\epsilon^2} \ln\left(\frac{1}{\delta + c_l}\right), \frac{|\mathcal{S}_h||\mathcal{A}_h|H_h^2}{\epsilon^2} \ln\left(\frac{1}{\delta + c_h}\right) \right) \right)$$

episodes, we will have $|V_o^{\pi_l^*, \pi_h^*} - V_o^{\pi_l^*, \pi_h}| + |V_o^{\pi_l^*, \pi_h} - V_o^{\pi_l, \pi_h}| \leq \epsilon$ and so $|V_o^{\pi_l^*, \pi_h^*} - V_o^{\pi_l, \pi_h}| \leq \epsilon$.

**Part (i)** Note that only learning the high-level policy when the low-level policy is optimal is equivalent to learning an $\epsilon$-accurate high-level policy interacting with $\mathcal{M}_h$ with a stationary transition function (since the low-level behaviour is not evolving anymore). Hence we can bound the number of episodes $N_h$ required to have: $|V_h^* - V_h^{\pi_l^*, \pi_h}| \leq \epsilon$, by directly applying Eq. (2) to the high-level MDP to get

$$\mathbb{E}[N_h] = \Omega\left( \frac{|\mathcal{S}_h||\mathcal{A}_h|H_h^2}{\epsilon^2} \ln\left(\frac{1}{\delta + c_h}\right) \right).$$

To be able to use this result to construct the bound of interest, we need to make sure these results can be translated into the original MDP: $|V_o^{\pi_l^*, \pi_h^*} - V_o^{\pi_l^*, \pi_h}| \leq \epsilon$. In particular, the reward functions are not the same for $\mathcal{M}_o$ and $\mathcal{M}_h$. We defined the original MDP's reward function to penalize each step unless it reaches the end goal, in which case it rewards the agent with a bonus. This reward function was initially defined to take values in the interval $[-a, b]$. Without loss of generality, we shift the reward function so that the rewards are now drawn from the interval $[-a - b, 0]$. In this new setting, a successful completion of the task is rewarded with a bonus of $0$. Similarly, the low-level reward function penalizes each step unless it reaches the instructed sub-goal. Lastly, the high-level reward function $r_h(s_h, a_h) = \sum_{t=1}^{H_l} r_l(s_{l,t}, a_h)$, consists of the sum of the low-level rewards accumulated during each sub-episode. A difference of scale in value function arises because high-level rewards include the bonus the low-level agent receives for completing each sub-goal, whereas the value functions in the original MDP do not include this bonus. To compensate for this difference, the low-level reward is re-scaled with a penalty twice larger; the low-level reward $r_l$ now takes values in $[-2a - 2b, 0]$, and, under this specific reward function, the completion of a sub-goal is rewarded by $0$ as well. The re-scaling ensures that the accumulated penalties are larger in the high-level MDPs, even for trajectories with a single intermediary step. This guarantees that $|V_o^{\pi_l^*, \pi_h^*} - V_o^{\pi_l^*, \pi_h}| \leq |V_h^* - V_h^{\pi_l^*, \pi_h}|$. Hence after $\mathbb{E}[N_h]$ episodes, we have $|V_o^* - V_o^{\pi_l^*, \pi_h}| \leq \epsilon$

**Part (ii)** By a similar argument to Part (i), we can bound the number of episodes in the low-level MDP required to obtain an $\epsilon$-optimal low-level policy for a fixed high-level policy $\pi_h$. In particular, a lower bound on the number of episodes $N_l$ required to have $|V_l^{\pi_l^*,\pi_h} - V_l^{\pi_l,\pi_h}| \leq \epsilon$ can directly be obtained from Eq. (2):

$$\mathbb{E}[N_l] = \Omega\left(\frac{|\mathcal{S}_l||\mathcal{A}_h||\mathcal{A}|H_l^2}{\epsilon^2}\ln\left(\frac{1}{\delta + c_l}\right)\right).$$

Ultimately, we are interested in a lower bound on the number of episodes in the original MDP $N_o$. Since a single episode in the original MDP corresponds to several episodes in the low-level MDP, we can divide the number of episodes by a factor $H_h$: $\mathbb{E}[N_o] \geq \frac{\mathbb{E}[N_l]}{H_h}$. However, the difference in episode length between the two MDPs also induces a difference of scale in their value functions. To ensure that the learned policies are $\epsilon$-optimal in the original MDP we need to ensure that $|V_l^{\pi_l^*,\pi_h} - V_l^{\pi_l,\pi_h}| \leq \frac{\epsilon}{H_h}$, which requires at least $H_h^2\mathbb{E}[N_l]$ low-level episodes. Combining the two arguments, we get that $\mathbb{E}[N_o] \geq \frac{\mathbb{E}[N_l]H_h^2}{H_h} = \mathbb{E}[N_l]H_h$. Since we are computing a lower bound on the number of episodes in the original MDP required to learn a near-optimal policy and recognising that $H_h \geq 1$, we can conclude that $\mathbb{E}[N_o] \geq \mathbb{E}[N_l]$.

This leads us to a lower bound on the number of episodes needed to obtain an $\epsilon$-accurate pair of policies as the one stated in the theorem. □

## A.2 Additional Experiments

### A.2.1 More $n$-rooms Problems

In the experimental section (Sec. 5), we used several room layouts. In the main paper, we only provide learning curves for mazes that are composed of rooms without any obstacles or mazes that are composed of all the possible room layouts depicted in the rightmost plot of Fig. 4. To complete our experiment, we show below in (Fig. 5 and Fig. 6) the learning curve obtained when mazes are built from two or three different room layouts. These learning curves highlight the same behaviour that we previously discussed. With simple mazes, the low-level MDP strongly dominates the bound but as we increase the high-level MDP complexity (i.e. the number of rooms), the benefit of the hierarchical machinery becomes evident. Note also that those results were used to plot the evolution of the bound ratio in the leftmost plot of Fig. 4.

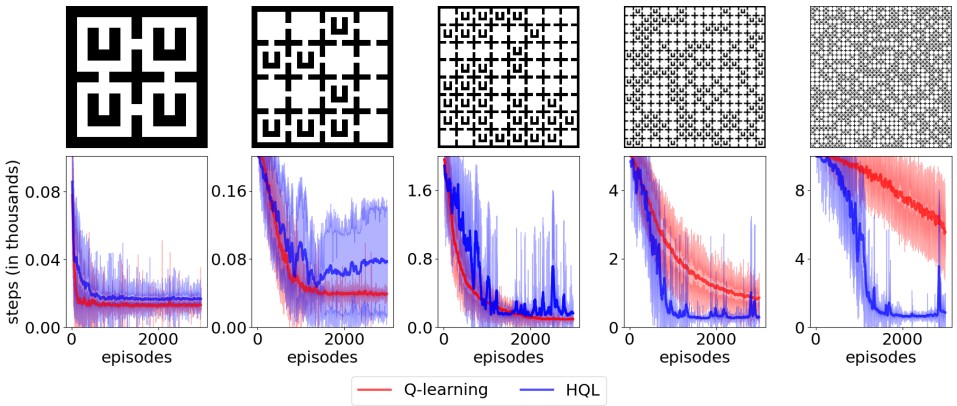

Figure 5: Shows learning curves on various maze sizes with two different room instances; either the room is empty, or it has a U-shape obstacle in it. The agent's performance is measured in the number of steps it requires to solve the task.

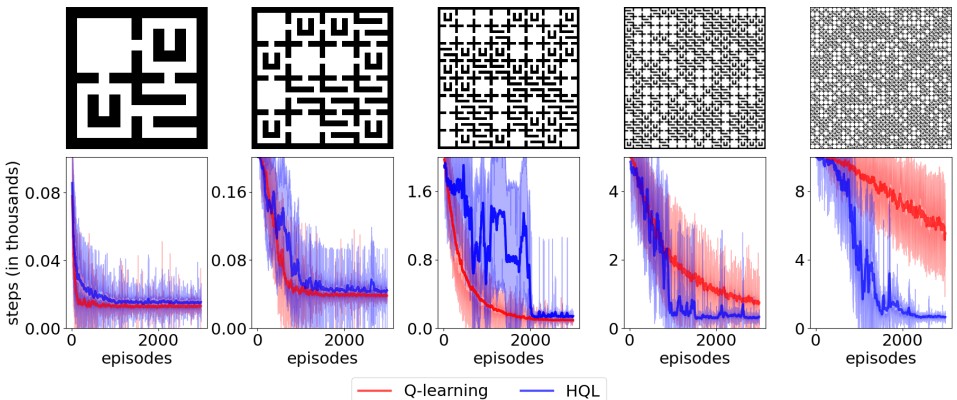

Figure 6: Shows learning curves on various maze sizes with three different room layouts; either the room is empty or it has either a U-shape obstacle or the room is stripped with horizontal walls. The performance of the agent is measured in the number of steps it requires to solve the task.

### A.2.2 Taxi Environment

Another task we consider is the taxi environment [9]. In this environment, the agent is tasked to pick up a passenger and bring them to a specific location. As illustrated in Fig. 7 the taxi environment is a grid world composed of 25 cells ($5 \times 5$ grid). At the beginning of an episode, the passenger is assigned a random initial location, which can only be in one of the four taxi stations depicted by the letters R, G, Y and B in Fig. 7 right plot, and a final destination which is any of the three remaining taxi stations. When visualising the environment, the location of the passenger is denoted by the letter in blue and the final destination by the purple letter. The initial position of the taxi is also randomly sampled, and to complete the task, the taxi will have first to pick up the passenger and then deliver him to his final destination.

This problem can be formalized by a MDP. The state space $\mathcal{S}$ encodes the 404 reachable states. While the 25 possible taxi locations, the 4 possible destinations and the 5 potential passenger locations suggest that there are 500 states. The states where the passenger is at the destination but the taxi is somewhere else are not possible and there are $24 * 4 = 96$ such configurations. The action space $\mathcal{A}$ consists of the six possible actions (move up, down, left, right, pickup, dropoff). The default horizon $H$ considered for this environment is 200 steps. The initial state distribution uniformly chooses the cell in which the taxi starts the episode as well as the passenger's initial location and destination. The reward function assigns a positive bonus (+20) for successfully delivering the passenger, a penalty for illegal action such as an inappropriate dropoff and pickup action (-10) and a small penalty (-1) for each remaining step. The characteristics of the original MDP for this problem are the following $|\mathcal{S}| = 404$, $|\mathcal{A}| = 6$ and $H = 200$.

To apply HSQL, the following decomposition was considered:
**(i) High-level MDP:** The high-level state space only encodes the current passenger location and the final desired destination, hence $|\mathcal{S}_h| = 20$. The high-level action space is composed of 8 possible sub-goals $|\mathcal{A}_h| = 8$. Each sub-goal encodes one of the four special locations and whether the passenger should be picked up or dropped off. Since all possible initial configurations can be completed with only two sub-goals, first a pick-up instruction and then a drop-off instruction, the horizon of the high-level MDP is $H_h = 2$.
**(ii) Low-level MDP:** The low-level agent needs to be able to perform all primitive actions so $\mathcal{A}_l = \mathcal{A}$. Then, the low-level state space encodes the current location of the taxi as well as the instructed sub-goal, yielding to a state space of size $|\mathcal{S}_l| = 200$. Since the trip is evenly divided into two separate instructions, the horizon of the low-level agent is $H_l = 100$.

By Theorem 3.1, we can observe that the low-level MDP is likely to dominate the bound and to drive the overall complexity of the problem. Ignoring the constant terms in the bound, our theory suggests that small improvement should still be possible. This is confirmed by our empirical results in Fig. 7 (left plot) which shows a significantly steeper learning curve for HSQL compared to Q-learning on the taxi task. In order to make the comparison between the two algorithms as fair as possible, we considered a range of possible hyperparameter values and for each algorithm, we only report the best-performing setting. The hyperparameters considered are the initial exploration rate

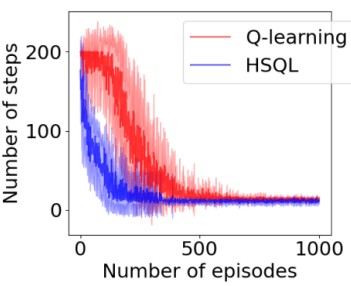 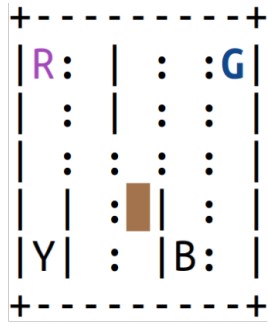

Figure 7: **Left plot** shows the learning curves of HSQL and Q-learning in the taxi environment. **Right plot** illustrates the taxi environment; the agent (yellow square) needs to navigate a grid world to reach the pickup location (encoded by the blue letter) and drop off the passenger at its final destination (encoded as the purple letter). The remaining letters are the possible pickup and drop-off locations. Note that the taxi needs to learn how to navigate the grid world in order to avoid the walls represented by solid lines.

$\epsilon \in [0.1, 0.3, \cdots, 0.7, 0.9]$ and the decay rate $\delta \in [1 - 10^{-3}, 1 - 10-4, \cdots, 1 - 10^{-7}]$. Note that we only considered the following decay function $\epsilon_{k+1} = \epsilon_k * \delta^{k+1}$, where $k$ denotes the current episode number. The error bars are obtained by running ten different seeds.

## B  The Option Framework and Goal-Conditioned HRL

As most of the existing theory on HRL [30, 12, 11] has been developed in the option framework, we recall below the definition of that framework and the connection it has with the goal-condition HRL framework considered in this article. An immediate difference is that those previous works consider an infinite horizon setting while we focus on a finite horizon setting. For the remainder of this section, we describe some other significant differences.

Let's first recall that the tuple $M = \langle \mathcal{S}, \mathcal{A}, p, r, \gamma \rangle$ describe a finite MDP, where $\mathcal{S}$ is a finite set of states, $\mathcal{A}$ is a finite set of actions, $p(s'|s, a)$ is the probability of transitioning to the state $s'$ given that in state $s$ action $a$ was executed, $r(s, a)$ is the reward distribution associated with the state action pair $(s, a)$ and, lastly, $\gamma$ is the discount factor. A policy $\pi : \mathcal{S} \to \mathcal{A}$ maps each state to an action. An option is a tuple $o = \{\mathcal{I}_o, \beta_o, \pi_o\}$, where $\mathcal{I}_o \subset \mathcal{S}$ is the initiation set, i.e. the set of states where the option $o$ can be started, $\beta_o : \mathcal{S} \to [0, 1]$ is the probability distribution to terminate in option in a given state and finally $\pi_o$ is the intra-option policy, i.e. the policy followed until the option ends.

Whenever the set of primitive actions $\mathcal{A}$ is replaced by a set of options $\mathcal{O}$ the original MDP M, becomes a semi-MDP $M_\mathcal{O} = \langle \mathcal{S}_\mathcal{O}, \mathcal{O}, p_\mathcal{O}, r_\mathcal{O}, \tau_\mathcal{O} \rangle$. The state space $\mathcal{S}_\mathcal{O} \subseteq \mathcal{S}$ is the set of all initial and terminal states. $\mathcal{O}$ denotes the set of options and $p_\mathcal{O}(s', o, s)$ is the probability to end option $o$ in state $s'$ given that it was started in state $s$:

$$p_\mathcal{O}(s', o, s) = \sum_{k=1}^{\infty} P(s_k = s'|s, \pi_o)\beta_o(s'),$$

where $P(s_k = s'|s, \pi_o)$ is the probability to reach state $s'$ in exactly $k$ steps from state $s$ under policy $\pi_o$. $r_o(s, o)$ is the distribution of reward cumulated when starting policy $\pi_o$ in state $s$. Lastly, $\tau_\mathcal{O}$ is the distribution of holding time, i.e. the number of primitive steps executed to reach $s'$ from $s$ under policy $\pi_o$.

It is possible to map the goal-conditioned HRL framework onto the options framework. Sub-goals would correspond to options, the initiation set of a given option $o$ includes all states where the corresponding sub-goal $g_{sub}$ is available, the option will terminate with probability 1 if the goal state has been reached or the $H_l$ time steps have been executed and terminate with probability 0 otherwise. Lastly, the intra-option policy $\pi_o$ corresponds to the low-level policy $\pi_l(\cdot|g_{sub})$ conditioned on the corresponding sub-goal.

From a theoretical standpoint, the key difference between the results in [11, 12, 30] and the ones presented in this article are that they assume the intra-option policies $\pi_o$ to be known. If we do not assume strong prior knowledge about the task, the number of potential intra-option policies will be prohibitive. In this article, we investigate algorithms that do not make assumptions about the low-level policy and jointly learn the low- and high-level policies through interactions with the environment, thus going beyond what has been done for the options framework.

