# OpenReview forum: "Sample Complexity of Goal-Conditioned Hierarchical Reinforcement Learning"
_NeurIPS.cc/2023/Conference — NeurIPS 2023 poster_

### Official Review · Reviewer_2sTV · 2023-07-05

**Soundness:** 3 good
**Presentation:** 3 good
**Contribution:** 3 good
**Rating:** 5
**Confidence:** 4

**Summary:**

The paper presents a theoretical analysis of the sample complexity in goal-conditioned hierarchical reinforcement learning (HRL) and establishes a lower bound using hierarchical decomposition to quantify it. Additionally, the paper empirically validates the theoretical results by examining the sample complexity of the proposed hierarchical algorithm on several toy grid-world tasks.

**Strengths:**

I commend the authors for addressing an important theoretical problem in the field of HRL and deriving a lower bound to quantify the sample complexity of goal-conditioned HRL.

**Weaknesses:**

One significant aspect that the paper lacks is a thorough theoretical analysis regarding the selection of sub-goal spaces in continuous environments or sets in discrete environments.

**Questions:**

Q1. The selection of the sub-goal space plays a vital role in the efficiency of the HRL algorithm, as different choices of sub-goal spaces can result in varying sample efficiencies, such as in HIRO [1], HRAC [2], HIGL [3], and others [4]. Unfortunately, in Theorem 3.1, the authors do not analyze the impact of different sub-goal spaces on the sample efficiency of HRL. Therefore, I find the theoretical results to be trivial.

[1] Nachum, Ofir, et al. "Data-efficient hierarchical reinforcement learning." Advances in neural information processing systems 31 (2018).

[2] T. Zhang, S. Guo, T. Tan, X. Hu and F. Chen, "Adjacency Constraint for Efficient Hierarchical Reinforcement Learning," in IEEE Transactions on Pattern Analysis and Machine Intelligence, vol. 45, no. 4, pp. 4152-4166, 1 April 2023.

[3] Junsu Kim, et al. "Landmark-guided subgoal generation in hierarchical reinforcement learning. " Advances in Neural Information Processing Systems, 34: 28336–28349, 2021.

[4] Lee, Seungjae, et al. "DHRL: A Graph-Based Approach for Long-Horizon and Sparse Hierarchical Reinforcement Learning." Advances in Neural Information Processing Systems 35 (2022): 13668-13678.

Q2. Another limitation of the paper is the absence of a comparison between the proposed method and other existing HRL algorithms in the experimental section. It would be valuable to include such a comparison to provide a more comprehensive evaluation of the proposed approach

**Limitations:**

See Questions

---

> ### Author Rebuttal · Authors · 2023-08-09
>
> We thank you for the time you dedicated to evaluating our work.
>
> We thank you for recognising that we are working on a “critical problem in HRL”. Our bound highlights the importance of the design of the subgoal space. In particular, the high level action space,which in our framework corresponds to the subgoal space, comes directly into the bound in equation 6. Additionally, the high-level action space's size impacts both the high-level and the low-level MDP. The aim of our analysis is to provide a tool that evaluates a given decomposition and relates the characteristics of that decomposition to the sample efficiency. Therefore, our bound provides a sanity check to determine whether a proposed decomposition could be beneficial. Investigating methods that learn a subgoal representation (or space) is beyond the scope of our paper, although it is a challenging problem of great interest so we will mention it as a future research direction.
>
> The lower bound we proposed is valid for any hierarchical algorithm; the point of our experimental validation was to show that the dynamics suggested by the lower bound manifest themself empirically. We proposed this algorithm to empirically observe some of the insights suggested by our lower bound. Of course, one could think of other algorithms for this problem, but that is beyond the scope of our work. The lower bound suggests that higher efficiency gains are obtained when the given decomposition balances as evenly as possible the complexity between the two MDPs (high- and low-level). This is observed empirically by our experiments that start with a high low-level complexity and as we increase the high-level complexity the improvement of the hierarchical method becomes more apparent.
>
> Best regards,
> The authors

---

> > ### Comment · Reviewer_2sTV · 2023-08-17
> > **Response**
> >
> > Thanks for the response. I maintain my stance that the omission of subgoal space characteristics (not only space size) in the theorems is a limitation of this study.  For instance, in the task that requires the agent to "take a key to unlock a door", if the subgoal space does not contain crucial subgoal: "key," then HRL will not learn an effective hierarchical policy. Note that this is not correlated with the size of subgoal space. Nonetheless, given the strong endorsement from the other reviewers for acceptance, I have adjusted my rating accordingly.

---

> > > ### Author Response · Authors · 2023-08-21
> > >
> > > We thank you for your time and for the fruitful feedback. We agree that the design of the sub-goal space is a particularly interesting and relatively unexplored problem. One point that we can clarify is that the family of decomposition we consider are "solvable", meaning that there exist a sequence of sub-goals that complete the task and that each sub-goal is solvable. This will be clarified in the paper, thanks for the suggestion. The design of the sub-goal space will also have repercussions on different parameters of the decomposition (e.g. the low-level horizon) which is somewhat captured by our lower bound in Theorem 3.1. However, understanding exactly how the sub-goal space influences these parameters and the corresponding lower bound remains an open question that is beyond the scope of the present paper.

---

> > > > ### Comment · Reviewer_2sTV · 2023-08-21
> > > > **Response**
> > > >
> > > > Thank you for your response. It's worth noting that the assumption of solvable decomposition is very strong, which should be highlighted more prominently within the main content of the text. Additionally, it would be valuable if the authors could delve into the limitation of their theorems, particularly in regards to their lack of consideration for the design aspects of the sub-goal space. Please discuss it in the conclusion section.  If these suggestions are incorporated, I would be inclined to give a rating of 5.

---

> > > > > ### Author Response · Authors · 2023-08-21
> > > > >
> > > > > Thank you, we will follow your recommendations. In our revised version we will mention that we assume solvable decompositions in the main text and we will also discuss limitations of the theorems regarding the design of the sub-goal space.

---

### Official Review · Reviewer_hRCv · 2023-07-06

**Soundness:** 3 good
**Presentation:** 3 good
**Contribution:** 3 good
**Rating:** 7
**Confidence:** 4

**Summary:**

The paper takes an important step toward quantifying the benefits achieved due to hierarchical decomposition of an MDP by deriving lower bound on sample complexity of goal-conditioned HRL algorithms. The paper also proposes a novel hierarchical Q-learning algorithm that exploits goal-based hierarchical decomposition of an MDP into a high-level and a low-level sub-MDPs and jointly learns their policies. The authors evaluate hierarchical policies for different decompositions on original MDPs to validate when decomposition provides benefits and whether it aligns with the derived bound.

The paper is well motivated and clearly written. The ideas of the paper to quantify benefits of hierarchical decomposition are novel. The derived theoretical guarantees on the lower bound are sound. I suggest clarifying that the derived bounds only apply to tabular setting of RL i.e. discrete state space problems in the Introduction. The theoretical findings are  backed by empirical results on maze environments of different sizes with convincing insights. The empirical evaluation would benefit from diversification of domains and tasks not restricted to navigation, and an investigation of bounds when using bad and good decompositions for the same environment.

**Strengths:**

(I) The paper provides strong theoretical guarantees on the lower bound of the number of episodes given a decomposition needed to learn an epsilon-accurate hierarchical policy, which also serves as a lower bound to learn an epsilon-accurate optimal policy.

(II) The paper also identifies properties relating to state and temporal abstractions and the size of the high-level action space from the derived bound that can improve sample efficiency.


**Weaknesses:**

(I) The assumptions regarding the scope of the derived bounds restricted to a tabular setting need to be clarified in the Introduction.

(II) All experiments are on maze environments of different sizes. While the current analysis is convincing for navigation tasks, it will be interesting to see if the benefits of decomposition and the derived bounds align for more diverse domains and tasks that not restricted to just navigation e.g. officeworld [2], taxiworld [3] etc.

(III) It is not clear how the bounds will identify when a decomposition is bad enough and would degrade the performance compared to non-hierarchical algorithms.

Minor errors:
(I) Line 48: proposes -> propose

**Questions:**

(I) How are the ideas of the Stationary Hierarchical Q-learning to overcome non-stationarity of the high-level policy related to the ideas in [1]?

(II) Can you elaborate what it means to separate the original state space evenly between the two level of hierarchy?

(III) Does the method apply only to dense reward functions?

(IV) Would the bounds identify when a decomposition for an environment would degrade performance of the proposed algorithm compared to Qlearning?

References:

[1] Levy, A., Konidaris, G., Platt, R. and Saenko, K., 2017. Learning multi-level hierarchies with hindsight. arXiv preprint arXiv:1712.00948.

[2] Rodrigo Toro Icarte, Toryn Klassen, Richard Valenzano, and Sheila McIlraith. Using reward machines for high-level task specification and decomposition in reinforcement learning. In International Conference on Machine Learning, pages 2107–2116. PMLR, 2018.

[3] Thomas Dietterich. State abstraction in maxq hierarchical reinforcement learning. Advances in Neural Information Processing Systems, 12, 1999.

**Limitations:**

(Included in the weaknesses)

---

> ### Author Rebuttal · Authors · 2023-08-09
>
> Thank you for your enthusiasm about our work and for your detailed feedback.
> In what follows we address the concerns raised during your review.
>
> i) “*The assumptions regarding the scope of the derived bounds restricted to a tabular setting need to be clarified in the Introduction.*”
>
> At the moment these assumptions are stated in the background section, but we agree with you, we could contextualise the work more thoroughly in the introduction allowing us to mention those assumptions. We will change this section accordingly.
>
> ii) “*It will be interesting to see if the benefits of decomposition and the derived bounds align for more diverse domains and tasks.*”
>
> While it would be interesting to investigate the performance on a different set of tasks, we choose to investigate maze-like environments because the hierarchical decomposition is natural and it is easy to modulate the complexity of the MDP i. e. adding more rooms or room layouts. The main purpose of the article is to better understand the relationship between the sample efficiency and the characteristics of the decomposition. To that end we focus on environments for which it is possible to modify at will the MDP’s complexity, hence yielding a diverse set of decomposed MDPs. This allows us to empirically verify the properties suggested by the derived lower bound and confirms that the introduction of the hierarchical machinery becomes more beneficial as the task’s complexity increases. That being said, we will integrate experiments on the taxi environment in the final version of the paper.
>
> iii) “*It is not clear how the bounds will identify when a decomposition [...] would degrade the performance compared to non-hierarchical algorithms.*” and “*Would the bounds identify when a decomposition for an environment would degrade performance of the proposed algorithm compared to Q-Learning?*”
>
> In theory, we can compare the HRL lower bound in Thm 3.1 and the RL lower bound given in eq. 4 to quantify if the problem could be solved more efficiently with an HRL algorithm. The idea is that if the HRL lower bound is larger than the one for standard RL, this may indicate that the decomposition is not necessarily helpful so we should proceed with caution. However, we should note that in practice, there is no guarantee that any proposed algorithm matches the lower bound efficiency. However, we agree that it would be interesting to propose a provably efficient algorithm for HRL. We will mention this as a potential future research direction in our conclusion.
>
> iv) “*How are the ideas of SHQL to overcome non-stationarity of the high-level policy related to the ideas in [1]?*”
>
> The idea used in [1] is similar to the one used for our algorithm, Stationary Hierarchical Q-learning. The idea is to leverage the fact that we know that an optimal low-level policy will reach the subgoal. In the off-policy case considered in [1] they use a form of hindsight correction, while in our on-policy scenario we have to filter the transitions on which we perform the update. For both the principle is the same: the non-stationary induced by the low-level changing behaviour is mitigated because high-level updates are computed using only transitions that exhibit a near optimal behaviour, but the difference is that we consider on-policy updates. We will make sure to refer to [1] when we discuss the update to highlight the similarity between the methods.
>
> v) “*Can you elaborate what it means to separate the original state space evenly between the two levels of hierarchy?*”
>
> State abstraction is an important concept in HRL that implies that, depending on its level, the policy needs different information about the current state. In the framework we consider, the state representation is decomposed (or separated), the high-level observes a part of the original state information and the low-level observes another part. For example, in our experiment, the high-level is only aware of the current room ID and the low-level only observes the current location within the room. As large state spaces are challenging for RL algorithms to explore, the state abstraction induced by the HRL framework represents an interesting opportunity to mitigate the issue of exploring a large state space. The “max” operator in equation 6 indicates that the overall HRL sample complexity is driven by the decomposed MDP (either low- or high-level MDP) with the largest sample complexity.  Hence, efficient decompositions will have to balance the complexity as evenly as possible between the two MDPs. The decomposition of the state space is then an important aspect to consider when finding this balance. However, our lower bound also highlights that other characteristics of the decomposition such as the subgoal space’s size and the respective time horizons have also an influence on the final sample complexity.  We will clarify the above in section 3, in the paragraph dedicated to the interpretation of the bound.
>
> vi) “*Does the method apply only to dense reward functions?*”
>
> We only assume that the reward is in an interval [-a,b] with a, b >= 0, this assumption is flexible enough to encode dense or sparse reward functions. So, our theory is also valid for sparse reward functions.
>
> Based on your feedback we propose the following list of the modifications:
> - Mention the tabular assumption in the Introduction.
> - Clarify the usage of the bound as explained above
> - Make correction on line 48: proposes -> propose
> - Add reference to [1] for the update “trick”.
> - Add experiments on the taxi domain.
> - Add clarification about the state space state decomposition.
> - Mention the design of provably efficient HRL algorithm as a future research direction.
>
> Best regards,
>
> The authors
>
> [1] Levy, A., Konidaris, G., Platt, R. and Saenko, K., 2017. Learning multi-level hierarchies with hindsight.

---

> > ### Comment · Reviewer_hRCv · 2023-08-17
> >
> > Thank you for your response. I've read all the responses and reviews, and I'd like to keep my rating the same.

---

> > > ### Author Response · Authors · 2023-08-21
> > >
> > > We thank you for your time and for the fruitful feedback.

---

### Official Review · Reviewer_VzQJ · 2023-07-07

**Soundness:** 3 good
**Presentation:** 4 excellent
**Contribution:** 4 excellent
**Rating:** 8
**Confidence:** 4

**Summary:**

Provides a sample bound on the complexity of goal-conditioned HRL algorithms based on the two MDPs they are decomposed into, and a Q learning algorithm to leverage these findings. Formulates HRL as a two-level problem, where the upper level passes actions to the lower level policy. The work proves a lower bound on the complexity of the HRL formulation which pivotally scales according to the size of the high level action space and the reusability of the low level space. A new algorithm is introduced which identifies the need for a consistent low level action space, and this method is asssessed in four-room gridworld domains.

**Strengths:**

This work provides a clean proof with a highly understandable sketch and a strong intuition. Together, this provides an extremely clear and easy-to-read description of the sample complexity of HRL In addition, the theory provides some clear insights into how to understand other HRL work.

This work provides a simple idea applied to the existing framework of HRL in the description of the high-level training based on low-level performance. It also seems like adding the intuition of the shared upper-level complexity term would be useful for keeping the size of the upper policy action space small (by somehow limiting the goals), which is empirically verified in other HRL work.


**Weaknesses:**

Figure 1 is difficult to comprehend, somehow managing to be simultaneously overly simple (this is a basic construction of hierarchical RL) and overly complicated (what is the intuition for the equations on the right-hand side?

This work contrasts against the Options framework in the first paragraph of the background, without specific Ying what the options framework is. As a framework itself, the options framework also makes no assumptions about prior knowledge, no prevents from state abstraction, both statements made about the framework.

This work spends much of the earlier part justifying the context of the goal-based hierarchy, but it appears that other than the state-based complexity term, there is no strict requirement that the hierarchy be goal based as opposed to simply parameter based. As long as there exists a measure of the performance of the lower-level policy, it seems like the same reasoning would apply.

The empirical results are somewhat lacking. In particular, while the proof should apply generally to HRL contexts, the work only empirically verifies in maze environments, and maze environments which are constructed to amplify the advantages of the upper-level policy. A different kind of environment such as a mountain car or multiple inverted pendulum would have been interesting, notwithstanding an environment that requires a deep RL method.

**Questions:**

See the weaknesses section

**Limitations:**

Limited empirical assessment in multiple domains

Additional evidence of how the terms of the bound translate to empirical results would be insightful

---

> ### Author Rebuttal · Authors · 2023-08-09
>
> Many thanks for your feedback and for your interest in our work. In the following, we try to clarify your concerns.
>
> Figure 1 is actually composed of two independent figures, the leftmost schema describes the HRL framework and the rightmost plot is simply an illustration of the MDP decomposition induced by the HRL framework. It can be seen that the “original” MDP can be decomposed into a high-level MDP and low-level MDP. This decomposition is described in great detail in section 2.2. We agree that the original caption of Figure 1 is a bit vague regarding the rightmost plot. We will modify the caption to explain more clearly what we are illustrating, and explicitly refer to section 2.2 where a complete description can be found.
>
> Thank you for pointing out this misleading statement about the option framework, we will make sure to introduce the option framework formally and correct the erroneous affirmation made about the framework.
>
> Regarding the additional experiment we agree with you that it would be interesting to investigate the performance on a different set of tasks. The reason why we investigated the maze environment first is simply because the hierarchical decomposition is natural in those environments and it is easy to modulate the complexity of the MDP i.e adding more rooms or adding more room layouts. Following your feedback, we will add experiments on the taxi environment to the final version of the paper.
>
> To summarise based on your comments we intend to do the following modification to the final draft:
> - Update caption of Figure 1.
> - Introduce more formally the option framework and remove erroneous claims
> - Add experiments based on the taxi environment.
>
> Best regards,
> The authors

---

> > ### Comment · Reviewer_VzQJ · 2023-08-15
> > **Response**
> >
> > We appreciate the response and believe that the contributions will improve the quality of the work

---

> > > ### Author Response · Authors · 2023-08-21
> > >
> > > We thank you for your time and for the fruitful feedback.

---

### Author Rebuttal · Authors · 2023-08-09

We are grateful for the time and the effort the reviewers dedicated to our work, which will strengthen our article. We are also pleased to see that several of the reviewers were enthusiastic about our work, especially on the writing: “clean proof with a highly understandable sketch and a strong intuition”, “provides an extremely clear and easy-to-read description of the sample complexity of HRL” and on the broader impact “provides some clear insights into how to understand other HRL work”.

Please find a response for each review question as a direct reply to the reviewer asking it.

Best regards,
The authors

---

### Decision · Program_Chairs · 2023-09-21

**Decision:**

Accept (poster)

**Comment:**

Hierarchical Reinforcement Learning (HRL) algorithms are attracting a lot of attention but their theoretical understandings are quite lagging behind. This paper offers some of the first sample complexity bounds for the proposed class of goal-conditioned RL algorithms, and uses it to design new RL algorithms. The reviewers are unanimous about accepting this paper.